# Fatty Acid Composition of Milk from Mothers with Normal Weight, Obesity, or Gestational Diabetes

**DOI:** 10.3390/life12071093

**Published:** 2022-07-21

**Authors:** Livia Simon Sarkadi, Miaomiao Zhang, Géza Muránszky, Réka Anna Vass, Oksana Matsyura, Eszter Benes, Sandor G. Vari

**Affiliations:** 1Department of Nutrition, Hungarian University of Agriculture and Life Sciences, 1118 Budapest, Hungary; zhang.miaomiao@phd.uni-mate.hu (M.Z.); geza.muranszky@lilichro.com (G.M.); 2Department of Obstetrics and Gynecology, University of Pécs Medical School, 7624 Pecs, Hungary; vass.reka@pte.hu; 3National Laboratory for Human Reproduction, University of Pécs, 7624 Pecs, Hungary; 4Department of Pediatrics No. 2, Danylo Halytsky Lviv National Medical University, 79010 Lviv, Ukraine; kaf_pediatrics_2@meduniv.lviv.ua; 5Department of Food and Analytical Chemistry, Hungarian University of Agriculture and Life Sciences, 1118 Budapest, Hungary; benes.eszter.luca@uni-mate.hu; 6International Research and Innovation in Medicine Program, Cedars-Sinai Medical Center, Los Angeles, CA 90048, USA; sandor.vari@cshs.org

**Keywords:** mother milk, breast milk, fatty acids, obesity, gestational diabetes

## Abstract

Gestation and the neonatal period are crucial periods in infant development. Many components of breast milk, including fatty acids, play an important role in strengthening the immune system. The aim of our research was to evaluate the fatty acid profiles of milk from 69 mothers, including subjects having a normal weight, obesity, or gestational diabetes. For the analyses, we used gas chromatography (GC) with flame ionization detection (FID) and GC coupled with mass spectrometry (GC/MS). The main fatty acids found in breast milk were palmitic acid (C16:0; 26–28%), linoleic acid (C18:2; 23–28%), and α-linolenic acid linoleic acid (C18:3; 15–17%), followed by myristic acid (C14:0; 5–8%), lauric acid (C12:0; 4–6%) and stearic acid (C18:0; 4–5%). The average breakdown of fatty acids was 50% saturated, 44% polyunsaturated, and 6% monounsaturated. Breast milk samples were classified using principal component analysis and linear discriminant analysis. Results showed that milk from the two major groups of obese and normal body mass index (BMI) could be distinguished with an accuracy of 89.66%. Breast milk samples of Hungarian and Ukrainian mothers showed significant differences based on the fatty acid composition, which variations are attributable to the mothers’ dietary habits.

## 1. Introduction

Human milk is an optimal source of nutrients for infants, as it contains essential nutrients, bioactive compounds, and immunological factors necessary for growth and development [1]. Milk contains irreplaceable compounds that are needed for ideal postnatal development, unlike most infant formulas, which lack bioactive factors [2,3,4].

Human milk is made up of about 87–88% water and solid components, including about 7% carbohydrates, 1% proteins, and 3.8% lipids [5,6,7]. Human milk fats are an important energy source and structural element and serve as regulatory factors [8]. The fat in breast milk fat is 98% triacylglycerols (TAGs), while the remainder is made up of phospholipids and cholesterol [9].

Fatty acids are the basic elements of milk fats and are essential for infant growth and development. They play important roles in the development of the central nervous system and in the induction of the immune response, besides having anticarcinogenic and antidiabetic effects [10,11].

The majority of fatty acids in human milk are saturated fatty acids (SFAs) [12] and monounsaturated fatty acids (MUFAs), followed by omega-3 (n-3) or omega-6 (n-6) polyunsaturated fatty acids (PUFAs), which play important biochemical roles. Long-chain polyunsaturated fatty acids (LCPUFAs) include linoleic acid (C18:2 n-6; LA), which is the precursor of arachidonic acid (C22:6 n-3; ARA); and α-linolenic acid (C18:3 n-3; ALA), which is the precursor of both eicosapentaenoic acid (C20:5 n-3; EPA) and docosahexaenoic acid (C22:6 n-3; DHA) [13,14]. The biosynthesis of LCPUFAs from LA and ALA involves the same set of enzymes, and, therefore, these processes compete with each other. A high LA supply inhibits endogenous DHA synthesis and results in higher circulating n-6 LCPUFAs, further limiting DHA incorporation into the developing brain, as the circulating n-3 and n-6 LCPUFAs also compete for brain uptake [15,16]. These essential fatty acids play important roles in infant visual, immune, cognitive, and motor development [17,18,19,20,21,22,23,24]. Since February 2022, all infant formulas marketed in the European Union must have a DHA content between 0.33% and 1.14% of total fat, without the minimum requirement for ARA [25].

Generally, the proportion of SFAs and MUFAs in human milk is relatively constant [26]. Long-chain saturated fatty acids in maternal milk, such as palmitic acid, originate either from the maternal diet or are released from maternal adipose tissue or liver metabolism [27,28,29,30]. Short-chain fatty acids such as caproic acid (6:0) and caprylic acid (8:0), as well as medium-chain fatty acids such as capric acid (10:0), lauric acid (12:0), and myristic acid (14:0), are synthesized de novo in the mammary gland [31,32]. Caproic acid, caprylic acid, capric acid, and lauric acid are linked to antimicrobial biological activities [30].

In naturally occurring unsaturated fatty acids, the configuration of double bonds is usually cis. Natural trans fatty acids (TFA) typically result from bacterial activity during rumen biohydrogenation of edible LA and ALA and are therefore found in ruminant-derived foods (milk and meat). Guillocheau et al. recently reviewed the body of scientific knowledge on natural TFAs such as trans-palmitoleic (trans-C16:1 n-7) and trans-vaccenic (trans-C18:1 n-7) acids and highlighted that they have similar biochemical characteristics (i.e., n-7 double bond) and physiological effects [33].

TFAs are believed to contribute to the development of certain diseases. When assessing the health effects of TFAs, a distinction should be made between trans fatty acids of natural and industrial origin. A high intake of industrial TFAs has negative effects on human health, such as an increase in the level of low-density lipoprotein (LDL-cholesterol, “bad cholesterol”) and a decrease in the level of high-density lipoprotein (HDL-cholesterol, “good cholesterol”), leading to increased risk of heart disease. Moreover, industrial TFAs increase the risk of atherosclerosis, besides inducing apoptosis and inflammation. However, some beneficial effects of natural (ruminant) TFAs have been reported [34].

Trans fatty acids have been detected in breast milk, primarily vaccenic acid (C18:1 n-7), and elaidic acid (C18:1 n-9), which are associated with the consumption of milk and dairy products [35]. Trans fatty acid intake in lactating women usually depends on their geographical location and socio-economic situation. Maternal TFA intake is important during pregnancy, as it correlates with fetal growth [36].

The fatty acid position in TAGs is related to the quality of human milk, as the sn-2 positioning of fatty acids plays an important role in fat digestion and absorption. The main SFA in human milk is palmitic acid (C16:0) [37], which is usually located in the middle (sn-2) position of TAG, thereby facilitating the action of pancreatic lipase. In addition, palmitic acid promotes the absorption of fat and calcium in the infant’s body. The sn-1 and sn-3 positions are occupied primarily by unsaturated fatty acids, of which approximately half is oleic acid (C18:1 n-9) [38].

A number of nutritional, environmental, and socio-economic factors influence the chemical composition of breast milk during lactation, including the stage of lactation, gestational age at delivery, maternal diet, and maternal metabolic diseases such as diabetes [39,40]. Numerous studies have shown that the mothers’ diet and health can greatly affect the fatty acid content of breast milk [41,42,43,44]. Accordingly, knowing more about the role and composition of key fatty acids in breast milk is essential for ensuring optimal infant nutrition [45].

Gestational diabetes (GD) is a growing medical problem, as over 200,000 cases are diagnosed per year, representing approximately 7% of all pregnancies [46].

GD is defined as glucose intolerance with onset or first recognition during pregnancy [47]. Although the exact cause of GD is still unclear, decreased maternal insulin sensitivity before pregnancy coupled with insufficient insulin response during pregnancy appear to be the main pathophysiological mechanisms underlying the development of GD [48].

An optimal concentration of fatty acid, especially that of LCPUFAs, in breast milk is very important. According to Wijendran et al. [49], infants of women with GD are born with DHA concentrations that are approximately one-half those of infants of women without GD. An earlier report showed that women with insulin-dependent diabetes mellitus produce breast milk with higher levels of medium-chain fatty acids and lower levels of LCPUFAs compared to healthy women [50].

Peila et al. recently reviewed the latest findings on the effects of GD on the composition of breast milk. Results from the 50 articles examined suggest that diabetes may alter the composition of human milk. However, there are still few studies in this area and the protocols used have some limitations; for example, the variability of only a few specific milk components was evaluated as markers for this syndrome [51].

The aim of our research was to assess differences in the fatty acid composition of breast milk samples by analyzing the milk of mothers with different health statuses (normal weight, obese, and/or GD). In addition, breast milk samples from different origins (Ukraine, Hungary) were also compared.

## 2. Materials and Methods

### 2.1. Human Milk Samples

Breast milk samples were collected from 60 mothers in Lviv, Ukraine, who represented four groups (15 mothers in each group): (i) normal body mass index (nBMI); (ii) obese (O); (iii) nBMI with GD (nBMI+GD); and (iv) O with GD (O+GD). Samples were also obtained from 9 mothers in Pecs, Hungary, who had nBMI.

Mothers who had given birth at the Danylo Halytsky Lviv National Medical University (Lviv, Ukraine) or at the University of Pécs (Pécs, Hungary) were recruited for our study. The research staff had informed the mothers about the procedures for breast milk collection and emptying the breast the day before milk samples were obtained. Samples collection was performed between the 10th and 12th postpartum weeks.

The study participants were asked to collect samples exclusively between 7 am and 11 am, a time period when breast milk has an average composition [2,3,4] three hours after they had emptied the same breast during the previous expression or three hours after the previous session of breastfeeding. For sample collection, the mothers were asked to completely empty the breast using a mechanical milk pump (Philips Avent, Farnborough, UK) into disposable sterile polypropylene tubes. From the total expressed volume, 2 mL were taken with a sterile enteral syringe and aliquoted into microtubes (Eppendorf, Hamburg, Germany). Following collection, milk samples were immediately placed in a freezer at −20 °C. Then the samples were shipped on dry ice to the laboratory and stored at −80 °C.

Maternal demographic information (including age, education, and ethnicity) was collected at the time of study recruitment, while clinical information (BMI, presence or absence of GD, and type of delivery) was obtained from hospital records. This information was used to evaluate possible relationships between milk samples and maternal characteristics.

### 2.2. Preparation of Fatty Acid Esters

The preparation of fatty acid methyl esters (FAME) was based on the method described by Muuse et al. [52]. The frozen breast milk samples were warmed to room temperature and homogenized in an ultrasonic bath for 1 min. A 0.2 mL volume of the milk sample was mixed with 1.8 mL of water, then 2 mL of n-hexane was added. The mixture was vortexed for 2 min, then centrifuge (2000 rpm, 5 min). A 1 mL volume of the hexane phase was mixed with 4 mL of methanolic Na-methylate solution (5% m/v) and vortexed for 4 min at maximum speed. The upper hexane layer was aspirated with a Pasteur pipette and vortexed twice with 1 mL of water for 1 min. After drying in Na_2_SO_4_, a sufficient amount (0.25 mL) was pipetted from the clear hexane solution for GC/MS analysis. Until analysis, all samples were stored frozen at −18 °C.

### 2.3. Gas Chromatography

GC with FID was used to analyze FAME, while GC/MS was applied to confirm the identification of components. The gas chromatographs used were a ThermoFinnigan Trace GC (Milan, Italy), AS 2000 sampler, split/splitless injector, FID detector with a BaseLine N2000 CDS Data system, and an HP 5890 Series II GC 7673 (Palo Alto, CA, USA), AS split/splitless injector, 5971 MSD with HP ChemStation software package and mass spectra libraries (Wiley 275 or NIST 05). The chromatographic parameters were: (i) column: SP2340, 30 m × 0.32 mm ID; (ii) injector: 220 °C split mode, split ratio 50; (iii) carrier: N2 flow 0.5 mL/min; (iv) temperature program: 70 °C hold 1 min, 140 °C rate 4 °C/min, 250 °C rate 1 °C/min; and (v) detector: FID, 280 °C. The FAME was identified by comparison of their relative retention times with authentic FAME standards (FAME MIX c4-c24, Sigma Aldrich 18919-1AMP, LRAC2656, Laramie, WY, USA). Mass distribution was calculated electronically using the quantification of peak areas with standard normalization. Data are expressed as a percentage of total fatty acids.

### 2.4. Statistical Analysis

The statistical analysis of participants’ data was performed using the Statistical Package for Social Sciences (SPSS) software (Ver23, IBM, Armonk, NY, USA). Continuous variables (expressed as mean ± SEM) were analyzed by ANOVA, while infant sex was analyzed using the chi-square test. A *p*-value < 0.05 was considered significant.

The fatty acid composition of the samples was submitted to descriptive statistics, and the differences among the groups were subjected to a one-way analysis of variance (ANOVA). Tukey’s post-hoc test with a *p* < 0.05 significance level was performed to confirm differences. Multivariate data analysis methods were used to comprehensively evaluate and visualize the fatty acid profile of breast milk samples. Principal component analysis (PCA), an unsupervised statistical method that focuses on the direction of maximum variation of the data, was used to analyze the biological patterns in the data set, with random leave-one-out (LOO) cross-validation for the samples from Hungarian and Ukrainian mothers with normal BMI. At the same time, discriminant analysis (DA) was used to classify Ukrainian samples based on the BMI and health conditions of the mothers. The analysis was performed separately on four groups (nBMI; nBMI+GD; O; O+GD) and on two groups (nBMI; O). LOO cross-validation was used for the validation of the models. The PCA was performed using the Unscrambler X 10.4 (CAMO, Oslo, Norway) software, while *t*-tests, ANOVA, and DA were conducted with SPSS software (Ver23, IBM, Armonk, NY, USA).

## 3. Results

The mothers involved in our study were Caucasians aged between 25 and 36 having no severe chronic medical conditions (Table 1). The maternal age and gestational age of newborns did not differ in the four groups. We found significant differences in the maternal BMI as obese mothers had significantly higher BMI (31.5 ± 0.6) than mothers with normal BMI (23.4 ± 0.3) and after the delivery and at the time of breast milk sample collection as well. BMI of obese mothers diagnosed with GD was highest (32.5 ± 0.5) compared to all other samples. Interestingly, they were more likely to give birth to male infants.

The total SFA in the breast milk of nBMI, nBMI+GD, O, and O+GD mothers was 48.38%, 47.53%, 42.82%, and 43.24%, respectively (Table 2). The main SFAs found in Ukrainian mother milk samples were palmitic acid (C16:0), followed by myristic acid (C14:0), lauric acid (C12:0), and stearic acid (C18:0). In addition, small amounts of caproic acid (C6:0), caprylic acid (C8:0), capric acid (C10:0), arachidic acid (C20:0), and eneicosanoic acid (C21:0) were detected in all samples, while lignoceric acid (C24:0) was found only in nBMI and nBMI+GD samples. The content of C12:0 was significantly (*p* < 0.05) lower in the O+GD group than in the other groups, the content of C14:0 was lower in women with normal BMI than in obese women, and C16:0 was lower in nBMI than in the other groups.

Of the MUFAs, oleic acid (C18:1, 3–4%), palmitoleic acid (C16:1; 2–3%) and eicosenoic acid (C20:1; ˂0.4%) were present (Figure 1). A trace amount of myristoleic acid (C14:1; ˂0.1%) was found only in nBMI+GD samples. The total MUFA content of breast milk was 6.39%, 6.31%, 5.88%, and 5.87% in the breast milk of women in the O+GD, O, nBMI, and nBMI+GD groups, respectively. The MUFA content of the groups did not differ significantly.

In terms of the LCPUFA composition of breast milk samples, LA and ALA were present in the highest amounts, accounting for 23–28% and 15–17% of total fatty acids, respectively (Figure 2). The total amount of all other LCPUFAs (C20:2, C24:2, C24:3, C20:4, C22:6) was less than 1.5%. Dihomo-gamma linolenic acid (C20:3 n-6; GLA ˂ 0.3%) was present only in the nBMI and nBMI+GD samples. The LA content of the breast milk from women of the nBMI+GD, O, and O+GD groups was significantly higher than that found in milk samples from the normal BMI groups.

The ratio of unsaturated and saturated fatty acids was 0.8 in normal BMI samples and 1.1 in breast milk samples from obese mothers.

We also compared the nBMI breast milk samples from two different countries, Ukraine and Hungary (Figure 3). Significant differences between breast milk samples were found in the amounts of caproic acid (H:0.02 ± 0.01% vs. U:1.66 ± 1.27%), palmitoleic acid (H:2.91 ± 0.67% vs. U:2.26 ± 0.28%), palmitic acid (H:32.44 ± 3.94% vs. U:25.80 ± 2.35%), LA (H:28.44 ± 2.36% vs. U:23.43 ± 2.28%), and ALA (H:9.33 ± 2.43% vs. U:15.20 ± 1.97%). In the Hungarian samples, palmitic acid, palmitoleic acid, and LA were higher than in the Ukrainian samples, while ALA was higher in Ukrainian samples than in Hungarian samples. The total levels of saturated (H:54 ± 11% vs. U:49 ± 11%) and monounsaturated (H:6.4 ± 1.5% vs. U:5.9 ± 1.4%) fatty acids were higher in the Hungarian samples, while polyunsaturated (H:38 ± 5% vs. U:40 ± 5%) fatty acids were higher in milk samples from Ukrainian mothers. GLA occurred only in the Ukrainian nBMI samples. These differences are probably due to differences in dietary habits.

### 3.1. Statistical Evaluation of Fatty Acid Composition of Ukrainian Breast Milk Samples

Given the health status of the mothers, we also performed a supervised classification with discriminant analysis (Figure 4). The results showed that the samples formed two distinct groups based on BMI. However, the samples of mothers with diabetes, especially the obese groups, could not be clearly distinguished based on this data set. According to the results, 58.6% of the cross-validated samples were correctly classified into four groups. Canonical function 1 was responsible for the discrimination of samples according to the BMI of mothers. Among the fatty acids, C16:0, C18:1, and C18:3 contributed the most to demarcating the samples, particularly for the obese group. In contrast, the effect of the C18:0 and C20:3 fatty acids shifted the samples toward the negative range. As shown in Figure 4, canonical function 2 may play a role in separating samples from healthy and nBMI+GD mothers. In this case, fatty acids C6:0, C20:2, C21:0, and C24:0 had the highest discrimination coefficients.

Based on the results of the discriminant analysis performed on the four groups, the analysis was also performed on two groups (Table 3), where the data set was split based only on the BMI of the mothers. The model accuracy, in this case, was 89.8% after cross-validation. The number of misclassified samples was six, five of which were from the nBMI group. This result also supports the finding that normal and obese breast milk samples can be distinguished from each other based on their fatty acid profile.

### 3.2. Comparison of Ukrainian and Hungarian Breastmilk Samples

The FA profile of nBMI breast milk samples from Ukraine and Hungary was analyzed by PCA. The variance of the data set was described at 79% by using PC1 (59%) and PC2 (20%) (Figure 5a). PC1 was responsible for the separation of samples based on their origin (Ukraine or Hungary). The C16:0, C18:2, and C18:3 fatty acids had the greatest effect on the classification, as confirmed by the *t*-test results, as there was a significant difference between the measured amounts of the three fatty acids in the two groups. It is also important to note the role of C20:2 and C20:3, as these were more characteristic of the Ukrainian samples. According to the loading plot (Figure 5b), the distance between samples along PC2 was most affected by the amount of C10:0, C12:0, C14:0, and C18:2 fatty acids.

## 4. Discussion

Breast milk is rich in water, protein, carbohydrates, lipids, vitamins, and minerals, plus numerous bioactive components—such as hormones, growth factors, enzymes, and live cells—to support the neonate’s healthy growth and development. The mother’s current diet determines the nutrients found in breast milk, as it mirrors the nutrients in the mother’s blood at any given time [53].

Half of a newborn child’s energy needs are met by lipids, which make up 3% to 5% of breast milk. Besides energy, lipids provide an important source of essential fatty acids and cholesterol. European women’s breast milk contains 35% to 40% saturated fatty acids, 45% to 50% monounsaturated fatty acids, and approximately 15% polyunsaturated fatty acids [54].

Maternal health greatly influences the chemical composition of breast milk and thus has a major impact on the future health of the baby [55].

Among the factors influencing the fatty acid composition of breast milk, the most frequently mentioned factors are maternal age, parity, duration of pregnancy, maternal BMI, maternal diet, stage of lactation, and daily breastfeeding rate, as well as the presence or development of diabetes during pregnancy and maternal nationality [10,14,32,42,56].

In our study, we compared breast milk samples from two countries, Ukraine and Hungary, in the normal BMI group to show how pregnant women’s fatty acid composition affects the fatty acid composition of breast milk during lactation.

Sinanoglou et al. [14] investigated the effect of maternal nationality, among other factors, on the fatty acid profile of mother’s milk samples. Total SFAs and the n-6/n-3 ratio in colostrum fat were significantly lower in Greek mothers as compared to the corresponding fat of mothers of different nationalities. In addition, colostrum fat from Greek mothers contained significantly higher oleic acid, eicosanoic acid, and MUFAs than colostrum from mothers of other nationalities. The average SFA and MUFA ratios of colostrum fat from Greek women (46.27% and 40.33%, respectively, of total fatty acids) were found to be in the European range (between 37.24% and 46.88% or 39.11% and 45.19%) [57]. Regarding our results, the average SFA and unsaturated fatty acid ratios in breast milk from nBMI Ukrainian women (48.37% and 40.30%, respectively, of total fatty acids) and in nBMI Hungarian women (53.90% and 38.31%, respectively, of total fatty acids) are slightly outside the European range. These differences are probably due to different dietary habits.

According to Bitman et al., a decrease in medium-chain fatty acids indicates a deterioration in fatty acid synthesis in the mammary gland, as well as increased oleic acid and polyunsaturated fatty acid concentrations, suggesting chain elongation [58]. Nasser et al. [30] found that high carbohydrate intake is likely to increase the ratio of C12:0 and C14:0 fatty acids in breast milk through the conversion of carbohydrates to medium-chain SFAs in the human breast. Due to the rapid metabolism and unique transport of medium-chain fatty acids, they have clinical advantages over long-chain fatty acids; in addition, medium-chain fatty acids play a vital role in the intestinal microbiota in neonatal physiology and immunity [59].

Our results confirm previous observations to the effect that there is a significant difference in the fatty acid composition of the breast milk of obese women compared to the milk of mothers with a normal BMI. Similar to our findings, most studies have shown that mother’s milk contains a high amount (34–47%) of SFAs. The predominant fatty acids were palmitic (about 20%), followed by stearic (about 6%) and myristic acids (about 4%) [60,61]. Many studies found that palmitic acid was positively correlated to BMI. Obese patients have increased de novo palmitic synthesis as well as lipoprotein lipase activity. As a result, lipoprotein lipase may selectively hydrolyze and store saturated versus mono- and polyunsaturated FAs in obese individuals [62].

High maternal body weight alters the nutrient content of breast milk; however, there are different views and analyses in the determination [63]. The differences are likely to be due to different dietary habits, sampling conditions, and the method used. Based on the health status of the mothers, we could separate two distinct groups based on BMI. We observed a significant (*p* < 0.05) decrease in lauric and myristic acid in the O and O+GD milk samples compared to the nBMI milk samples, while levels of medium-chain fatty acids, such as capric and lauric acid, increased in nBMI breast milk. This finding was consistent with other findings found in the literature [61,64].

Regarding the MUFA content, it was found that most MUFAs remain stable over the course of lactation, with the exceptions of gondoic (C20:1), erucic (C22:1), and nervonic (C24:1) acids that decreased over time [30]. Similar to our results, C16:1, C18:1, and C20:1 were identified as the three main MUFAs in human milk, with a percentage of around 3–4% [61]. The total MUFA content in our samples was approximately 6% and did not show significant change between groups.

LCPUFAs play an important structural and regulatory role in the body, as there is evidence that n-3 fatty acids promote normal mitochondrial function and reduce excitotoxicity [65]. Human studies have demonstrated the effects of EPA and DHA in the resistance or treatment of central nervous system injuries or disorders, such as stroke [66] and epilepsy [67]. DHA and ARA treatment resulted in a decreased IL-1-induced pro-inflammatory response in human fetal and adult intestinal epithelial cells [68]. The effect of LCPUFAs on gut barrier function may explain some of their protective effects in intestinal inflammatory disease [69]. LCPUFAs are also known to play an important role in the development and function of the immune system [70,71].

LCPUFAs are usually present as a minor constituent of mother’s milk [13]. Based on the data analysis by Floris et al. [60], it was found that several LCPUFAs seemed to decrease in both preterm and term milk, including DHA and docosapentaenoic acid (C22:5; DPA). GLA showed a steady rise between transitional and mature milk, while linoleic acid, EPA, and ALA remained relatively stable. Isesele et al. [72] found that obese women had higher levels of LA than non-obese women. LA is the precursor for AA, and it is converted to AA through a series of delta-6 desaturation, elongation, and delta-5 desaturase reactions [73]. Obesity has been linked to increased delta-6 desaturase activity in numerous studies [74,75]. A significant positive correlation has been found between LA intake and LA levels in human milk [35].

In our study, nBMI breast milk samples had significantly lower (23.8%) LA levels than the nBMI+GD (25.8%), O (26.8%), and O+GD (27.8%) samples. Similar to Isesele et al. [72], we found no significant differences in AA levels between the normal and obese BMI groups, implying that maternal obesity has little effect on AA levels. However, a small amount of GLA metabolite (˂0.3%) was found in the nBMI and nBMI+GD breast milk samples.

It has long been known that the number of insulin receptors in the mammary gland increases during lactation, and insulin appears to be involved in the physiological regulation of lipogenesis [76]. Insulin sensitivity affects the action of enzymes needed for the synthesis of long-chain fatty acids, such as elongases and desaturases. Saturated and very long-chain fatty acids are characteristic of biochemical abnormalities such as gestational diabetes, while palmitic acid is associated with impaired insulin resistance markers and the risk of GDM. Zhu et al. [77] found that high even-chain and low odd-chain SFA concentrations may have a synergistic effect associated with an increased risk of subsequent GDM, implying that both types of SFA may play a role in glucose homeostasis. The combined effects of plasma phospholipids and SFA subclasses, as well as endogenous biosynthetic pathways, complicate the interpretation of SFA composition in relation to disease risk.

In a recently published paper, Covaciu et al. [78] underlined lactation stages as a function of fatty acid and elemental contents in order to identify the best discriminant markers. According to our results, 58.6% of the cross-validated samples were correctly classified into four groups: women with a normal BMI, women with a normal BMI and GD, obese women, and obese women with GD. The highest contribution to the separation and differentiation of the sample’s fatty acids was observed for C16:0 and C18:1, while C18:3 fatty acids were characteristic in the obese group. In the nBMI and nBMI+GD groups, mother’s milk arachidic, palmitic and tetracosanoic fatty acids had the highest discrimination coefficients.

The most important limitation of our study is that we lacked detailed information about the dietary habits of the pregnant women and lactating mothers who participated in our study. Recently published reports underline the importance of maternal nutrition during pregnancy, as it can affect the metabolism of the fetus and the health of the neonate.

Although much research has been performed in the area of healthy nutrition for infants, much work remains to be done in investigating the composition of breast milk [79].

## 5. Conclusions

The main fatty acid composition (palmitic, oleic, and linoleic acids) showed only modest differences among the breast milk of mothers with normal BMI, normal BMI with GD, and obesity or obesity with GD. In contrast, minor fatty acids (caproic, caprylic, lauric, and myristic acids) showed greater divergences among these groups. LCPUFAs occurred in significantly higher amounts in milk from obese mothers compared to normal-weight mothers.

In further studies, it will be important to examine nutritional and health factors, besides breast milk handling techniques that may influence the fatty acid composition of mother’s milk, as well as the role of fatty acids in enhancing the immune system and preventing disease.

## Figures and Tables

**Figure 1 life-12-01093-f001:**
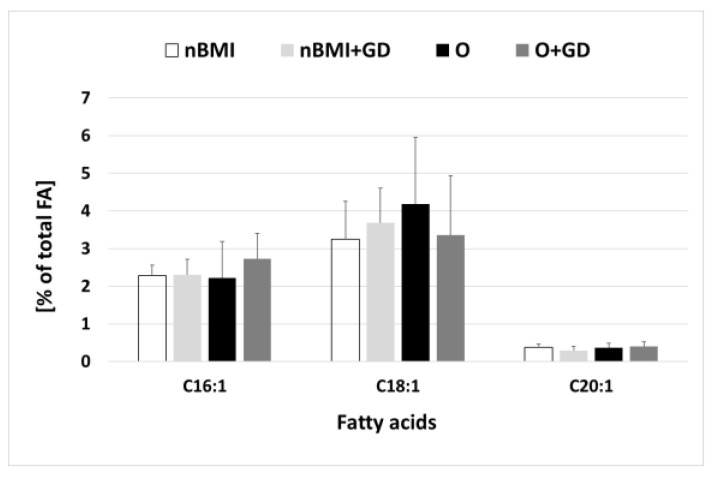
Monounsaturated fatty acids in breast milk samples from Ukraine. Total number of samples: 58, 15 per group (nBMI. nBMI+GD), 14 per group (O, O+GD). FA—fatty acid; nBMI—women with normal BMI; nBMI+GD—women with normal BMI and GD; O—obese women; O+GD—obese women with GD; C16:1—palmitoleic acid; C18:1—oleic acid; C20:1—eicosenoic acid.

**Figure 2 life-12-01093-f002:**
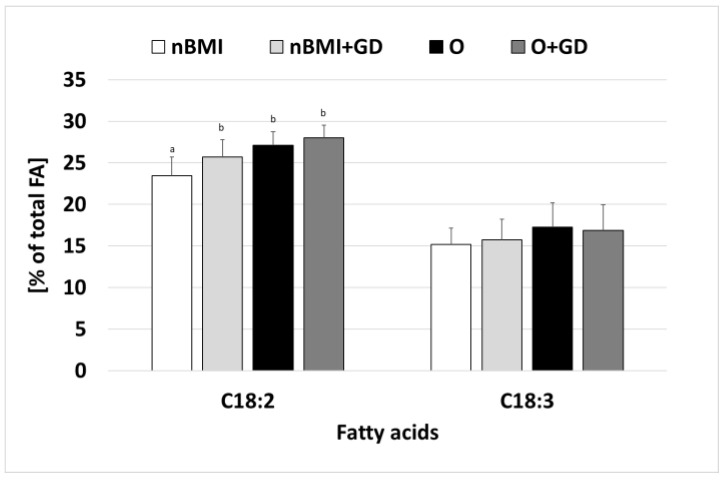
Long-chain polyunsaturated fatty acids in breast milk samples from Ukraine. Total number of samples: 58, 15 per group (nBMI. nBMI+GD), 14 per group (O, O+GD). FA—fatty acid; nBMI—women with normal BMI; nBMI+GD—women with normal BMI and GD; O—obese women; O+GD—obese women with GD; C18:2—linoleic acid; C18:3—α-linolenic acid; different letters show significant differences at *p* < 0.05 level.

**Figure 3 life-12-01093-f003:**
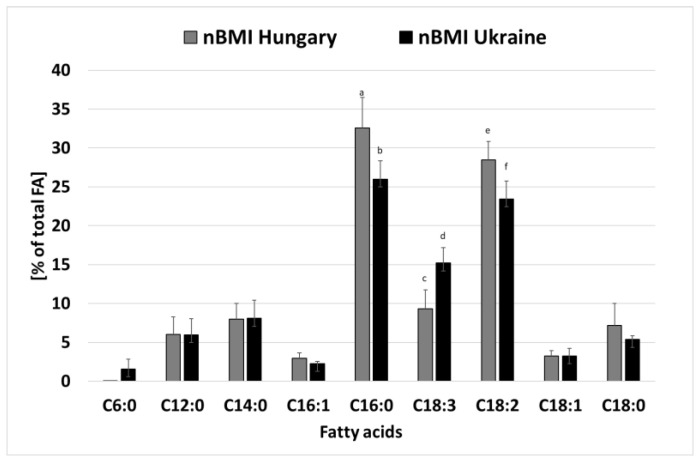
Main fatty acids in breast milk of nBMI women from Ukraine and Hungary. FA—fatty acid; nBMI—women with normal BMI; C6:0—caproic acid; C12:0—lauric acid; C14:0—myristic acid; C16:0—palmitic acid; C18:0—stearic acid; C16:1—palmitoleic acid; C18:1—oleic acid; C18:2—linoleic acid; C18:3—α-linolenic acid; different letters show significant differences at *p* < 0.05 level.

**Figure 4 life-12-01093-f004:**
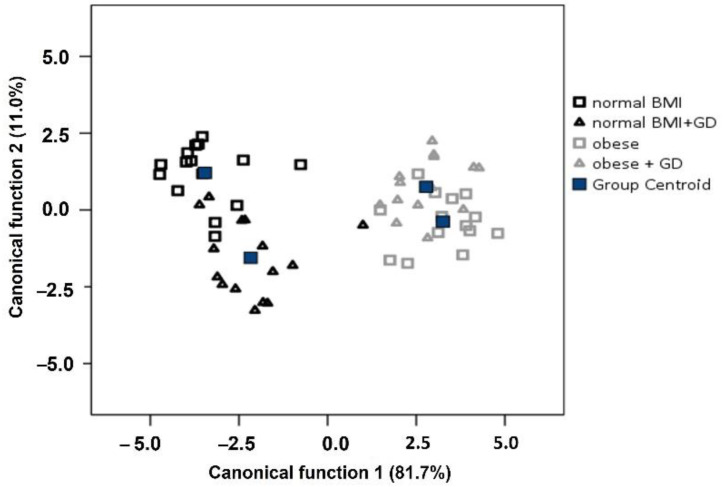
Discriminant analysis of Ukrainian breast milk samples.

**Figure 5 life-12-01093-f005:**
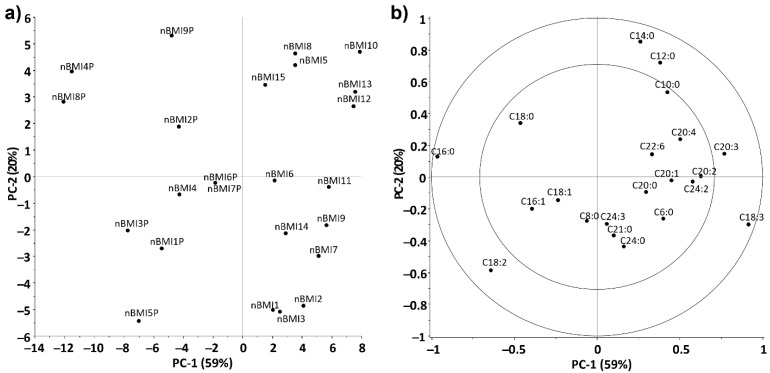
(**a**) Scores plot of principal component analysis of the fatty acid profile of breast milk samples from women with normal BMI in Hungary (P) and Ukraine (triangles: Hungary; dots: Ukraine). (**b**) Principal component loadings of the fatty acid data of milk samples from women with normal BMI in Hungary and Ukraine. C6:0, caproic acid; C8:0, caprylic acid; C10:0, capric acid; C12:0, lauric acid; C14:0, myristic acid; C16:0, palmitic acid; C18:0, stearic acid; C20:0, arachidic acid; C21:0, heneicosylic acid; C24:0, lignoceric acid; C16:1, palmitoleic acid; C18:1, oleic acid; C20:1, eicosenoic acid; C18:2, linoleic acid; C20:2, eicosadienoic acid; C18:3, α-linolenic acid; C20:3, dihomo-gamma-linolenic acid; C20:4, arachidonic acid; C22:6, docosahexaenoic acid. Total number of samples was 24.

**Table 1 life-12-01093-t001:** Data of participants.

	Normal BMI *	Normal BMI * with GD	Obese	Obese with GD
Number of involved mothers	24	15	15	15
Maternal age (years)	27.1 ± 1.2	27.8 ± 1.3	26.3 ± 1.2	30.7 ± 1.8
Gestational age (weeks)	39.5 ± 0.6	38.2 ± 0.2	37.8 ± 0.2	39.6 ± 0.4
Maternal BMI at delivery	23.4 ± 0.3 ^a^	23.1 ± 0.4 ^a^	31.5 ± 0.6 ^b^	32.5 ± 0.5 ^c^
Sex of neonate				
Female	11	8	10 ^c^	4 ^d^
Male	13	7	5 ^d^	11 ^c^
Delivery				
Natural	16	8	14	11
Cesarean section	8	7	1	4
Maternal education level				
Primary school	1	1	2	2
High school	10	12	7	9
College and above	13	3	6	4

* BMI: body mass index; GD: gestational diabetes; significant difference is indicated by different letters (*p* < 0.05).

**Table 2 life-12-01093-t002:** Saturated fatty acid composition of breast milk samples from Ukraine (% of total FA).

Fatty Acid	Normal BMI (*n* = 15)	Normal BMI+GD (*n* = 15)	Obese (*n* = 14)	Obese+GD (*n* = 14)
C6:0	1.66 ± 1.71 ^a^	0.17 ± 0.42 ^b^	0.24 ± 0.46 ^c^	0.11 ± 0.27 ^d^
C8:0	0.22 ± 0.53	0.04 ± 0.09	0.01 ± 0.04	0.07 ± 0.11
C10:0	1.05 ± 0.62	0.84 ± 0.21	0.79 ± 0.22	0.78 ± 0.30
C12:0	5.75 ± 2.20 ^b^	5.25 ± 1.99 ^b^	5.04 ± 1.30 ^b^	3.57 ± 2.00 ^a^
C14:0	7.74 ± 2.54 ^b^	7.14 ± 3.44 ^b^	4.13 ± 3.82 ^a^	4.58 ± 4.70 ^a^
C16:0	25.80 ± 2.21 ^a^	28.64 ± 2.78 ^b^	27.85 ± 2.75 ^b^	29.09 ± 3.45 ^b^
C18:0	5.28 ± 0.55	5.31 ± 1.70	4.57 ± 1.66	4.86 ± 1.42
C20:0	0.13 ± 0.05	0.08 ± 0.05	0.17 ± 0.16	0.13 ± 0.09
C21:0	0.41 ± 1.04	0.01 ± 0.03	0.02 ± 0.03	0.05 ± 0.11
C24:0	0.33 ± 0.69	0.06 ± 0.13	nd	nd
∑SFA	48.37 ± 1.63	47.53 ± 2.17	42.82 ± 3.12	43.24 ± 2.48

BMI: body mass index; GD: gestational diabetes; caproic acid (6:0); caprylic acid (8:0); capric acid (10:0); lauric acid (12:0); myristic acid (14:0); palmitic acid (16:0); stearic acid (C18:0); arachidic acid (C20:0); eneicosanoic acid (C21:0); lignoceric acid (C24:0); ∑SFA: total saturated fatty acids; different letters ^(a,b,c,d)^ show significant differences at *p* < 0.05 level; n: number of samples; nd: not detected.

**Table 3 life-12-01093-t003:** Results of discriminant analysis of two groups based on the BMI of mothers.

			Predicted Group Membership	
			Normal BMI	Obese
Original	Count	Normal BMI	29	1
		Obese	0	28
	%	Normal BMI	96.7	3.3
		Obese	0.0	100.0
Cross-validated	Count	Normal BMI	25	5
		Obese	1	27
	%	Normal BMI	83.3	16.7
		Obese	3.6	96.4

Original: results of calibration; cross-validated: results after leave-one-out cross-validation.

## Data Availability

The data presented in this study are available on request from the corresponding author.

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
