# Peer review of "Fatty Acid Composition of Milk from Mothers with Normal Weight, Obesity, or Gestational Diabetes"

_life, 2022, doi:10.3390/life12071093_

Round 1

Reviewer 1 Report

Although the manuscript has been improved, compared to the previous version, in my opinion, it still has a lot to improve before being acceptable for publication.

The authors have added some new results and statistical analyses.

However, the description and the interpretation of both the results and the statistical analysis is very poor.

In addition, they did not find statistical differences between groups, or at least they do not describe them. The reason for the lack of differences in my opinion is, most likely, due to the method used to analyze fatty acids, as I commented in the previous review. The method used in the work only allows determining the major fatty acids. It does not allow to differentiate, for example, between cis and trans isomers, nor to detect CLA. This is a major drawback, because differences in these minor FA may be the key to differentiating between groups.

Even so, the inexistence of differences in the main FA between groups is also an acceptable result, but its scientific significance is low.

The discussion of the results remains being very poor. They have added some sentences, that, in my opinion, do not improve the discussion, because they talk about their own results, than, other other hand, have not been reported in the results section.

More specific comments

Introduction:

Line 32: …source of nutrientes…

Line 35. ….bioactive factors…

Line 48:  it can only create double bonds at the n-9 or previous positions from the methyl end (n-9 desaturase is not the only desaturase in mammals).

Line 49:  Linoleic and linolenic acids are essential. Mammals can elongate and add double bonds to these FA, as authors comment later, and synthesize other PUFA.

The paragraph between lines 50-56 should go later, after the paragraph that finishes in line 90.

The paragraph between lines 64-73 should go before, after the paragraph that finishes in line 49. 

Line 78:  What do authors mean with "common.. effects"?

Line 89-90: The statement “From February 2022, all infant formulas marketed in the European Union must contain 0.33-1.14% of total fat DHA, without the minimum requirement for ARA [37]” should go at the end of the paragraph that talks about PUFA (line 72).

Lines 80-88:  As I commented in the previous review,  not all trans fatty acids are associated with adverse health effects. It is important to differentiate between trans fatty acids of industrial origin and those of ruminant origin. Although I raised many comments on this paragraph, the authors have not changed it at all. They just added a final sentence not referring to trans FA but to PUFA which should be moved to another paragraph.

Lines 115-117: General knowledge, it should be deleted, in my opinion.

Results: 

Table 1: the way to express significant differences between groups is confusing. It should be more appropriate to use different superscripts in the same row to indicate them, and explain the meaning in the base of the table. For example,

 Maternal BMI at delivery    23.4 ± 0.3a     23.1 ± 0.4a    31.5 ± 0.6b  32.5 ± 0.5b

 On the other hand, I do not understand the statistics applied to the gender of the newborn.

Lines 201-239: The authors do not report any statistical difference in the proportions of individual FA between the groups. It is not clear if it is because there is no difference or because they have not reported it. If there is any significant difference, they have to report it and they should indicate it on the figures. If they did not find any significant difference, they should state it in the text. 

Lines 222-223: The statement “The total MUFA content of breast milk in women of the obese group was a little higher (6.7%) compared to milk samples from the normal BMI groups (6.1%)” can be said if the statistical analysis has been carried out giving a significant difference. If this is the case, standard deviation values must be added to each data. If not, the sentence should be removed.

The same must be applied to the content of PUFA (line 237).

Lines 266-270: The interpretation of the PCA, in my opinion, is very poor.

Figures 4 and 5 should be put together, as figures 4a and 4b. In this way, it could be observed which FAs are related to each group. In any case, I do not agree with the interpretation made by the authors, because the points in figure 5 appear very scattered and there is no clear differentiation between groups. This is in agreement with the fact that there is no significant differences in the proportion of FA between groups, at commented before.

Lines 279-283 and Figure 6: I do not understand what is represented in figure 7. Authors should explain it better. On the other hand, author must give the discriminant equation, especially, which FA are included in it.

In my opinion, PCA and LDA are two statistical analyses that have similar objectives. Therefore, including both is redundant. Therefore, authors should include only the one that gives the clearest results.

Lines 305-309: As commented before, differences in proportion of individual FA and FA groups between mother groups can be stated only if the proper statistical analysis has been carried out, and SD values have to be added. They cannot be inferred from the PCA.

Figures 7 and 8 should be put together, as figures Xa and Xb. In this way, as commented before, it could be observed which FAs are characteristic of each group.

Discussion

Line 328:  We observed a decrease in medium-chain fatty acids (capric, lauric and myristic acid) and an increase in oleic acid in the O and O+GD milk samples”. This has not be reported in results, no statistical differences were reported among groups for these FA.

Author Response

Answers to Reviewer 1’s comments

Thank you for your valuable comments. We improved our manuscript taking into account all your suggestions.

We have inserted our answers below your comments

Comments and Suggestions for Authors

Although the manuscript has been improved, compared to the previous version, in my opinion, it still has a lot to improve before being acceptable for publication. The authors have added some new results and statistical analyses. However, the description and the interpretation of both the results and the statistical analysis is very poor.

In addition, they did not find statistical differences between groups, or at least they do not describe them. The reason for the lack of differences in my opinion is, most likely, due to the method used to analyze fatty acids, as I commented in the previous review. The method used in the work only allows determining the major fatty acids. It does not allow to differentiate, for example, between cis and trans isomers, nor to detect CLA. This is a major drawback, because differences in these minor FA may be the key to differentiating between groups.

Answer – We understand your criticism, but as we mentioned our previous answer the method commonly used for the determination of fatty acids was chosen for the study. No attempt was made to develop a method to distinguish cis and trans isomers in this study. The goals were to investigate the disease or health condition related changes and that we demonstrated. As part of the original call for Life Special Issue, The Effects of Nutrition on Pregnancy on the Mother and the Newborn, we wanted to published an observational research article rather than analytical chemical research and development work.

CLA has unique physical properties and a number of possible positional and geometric isomers, making lipid analysis a challenge. In our further study will also focus on the determination of conjugated linoleic acid (CLA) and vaccenic acid (VA) in the milk of mothers consuming different diets. We are also convinced that this comparison is an important scientific result. There are some good examples in the literature e.g.

(i) Martysiak-Żurowska D, Kiełbratowska B, Szlagatys-Sidorkiewicz A. The content of conjugated linoleic acid and vaccenic acid in the breast milk of women from Gdansk and the surrounding district, as well as in, infant formulas and follow-up formulas. nutritional recommendation for nursing women. Dev Period Med. 2018;22(2):128-134. doi: 10.34763/devperiodmed.20182202.128134. PMID: 30056399; PMCID: PMC8522894.

(ii) Sánchez-Hernández S, Esteban-Muñoz A, Giménez-Martínez R, Aguilar-Cordero MJ, Miralles-Buraglia B, Olalla-Herrera M. A Comparison of Changes in the Fatty Acid Profile of Human Milk of Spanish Lactating Women during the First Month of Lactation Using Gas Chromatography-Mass Spectrometry. A Comparison with Infant Formulas. Nutrients. 2019 Dec 14;11(12):3055. doi: 10.3390/nu11123055. PMID: 31847315; PMCID: PMC6950188.

Even so, the inexistence of differences in the main FA between groups is also an acceptable result, but its scientific significance is low.

Answer – In our opinion, the scientific significance of life science research depends on the useful information it provides to the patient, in our case a pregnant woman, and to physicians. We believe that our results have contributed to the expansion of knowledge in its current form. In our further research, we will also focus on the comprehensive study of fatty acids, which provides an opportunity to clarify deeper relationships.

The discussion of the results remains being very poor. They have added some sentences, that, in my opinion, do not improve the discussion, because they talk about their own results, than, other hand, have not been reported in the results section.

Answer – Additional information has been added to the discussion section (marked in color)

More specific comments

Introduction:

Line 32: …source of nutrientes…

Answer – corrected

Line 35. ….bioactive factors…

Answer – corrected

Line 48:  … it can only create double bonds at the n-9 or previous positions from the methyl end (n-9 desaturase is not the only desaturase in mammals).

Answer – The criticized sentence has been deleted

Line 49:  Linoleic and linolenic acids are essential. Mammals can elongate and add double bonds to these FA, as authors comment later, and synthesize other PUFA.

Answer – The criticized sentence has been deleted

The paragraph between lines 50-56 should go later, after the paragraph that finishes in line 90.

Answer – the proposed change has been done (88-94)

The paragraph between lines 64-73 should go before, after the paragraph that finishes in line 49. 

Answer – the proposed change has been done  (Line 47-59)

Line 78:  What do authors mean with "common.. effects"?

Answer –  the word "common" was changed to “similar” and the sentence was amended as follows: …. they have similar biochemical characteristics (i.e. n-7 double bond) and physiological effects (Line 72-73)

Line 89-90: The statement “From February 2022, all infant formulas marketed in the European Union must contain 0.33-1.14% of total fat DHA, without the minimum requirement for ARA [37]” should go at the end of the paragraph that talks about PUFA (line 72).

Answer – the proposed change has been done

Lines 80-88 As I commented in the previous review, not all trans fatty acids are associated with adverse health effects. It is important to differentiate between trans fatty acids of industrial origin and those of ruminant origin. Although I raised many comments on this paragraph, the authors have not changed it at all. They just added a final sentence not referring to trans FA but to PUFA which should be moved to another paragraph.

Answer – The information on trans fatty acids has been clarified and a new reference has been included (Line 74-82)

Lines 115-117: General knowledge, it should be deleted, in my opinion.

Answer – text delated

Results: 

Table 1: the way to express significant differences between groups is confusing. It should be more appropriate to use different superscripts in the same row to indicate them, and explain the meaning in the base of the table. For example,

 Maternal BMI at delivery    23.4 ± 0.3a     23.1 ± 0.4a    31.5 ± 0.6b  32.5 ± 0.5b

 On the other hand, I do not understand the statistics applied to the gender of the newborn.

Answer – Continuous variables (expressed as mean ± SEM), were analyzed by ANOVA while infant gender was analyzed using the chi-square test. A p value <0.05 was considered significant. Thank you for your suggestion, we applied superscripts to show significant differences.

Lines 201-239: The authors do not report any statistical difference in the proportions of individual FA between the groups. It is not clear if it is because there is no difference or because they have not reported it. If there is any significant difference, they have to report it and they should indicate it on the figures. If they did not find any significant difference, they should state it in the text. 

Lines 222-223: The statement “The total MUFA content of breast milk in women of the obese group was a little higher (6.7%) compared to milk samples from the normal BMI groups (6.1%)” can be said if the statistical analysis has been carried out giving a significant difference. If this is the case, standard deviation values must be added to each data. If not, the sentence should be removed.

The same must be applied to the content of PUFA (line 237).

Answer – Our response to some of the comments on statistics above is that we have performed the statistical analysis and modified the description accordingly.

Lines 266-270: The interpretation of the PCA, in my opinion, is very poor.

Answer –  A more detailed description was provided for PCA analysis

Figures 4 and 5 should be put together, as figures 4a and 4b. In this way, it could be observed which FAs are related to each group. In any case, I do not agree with the interpretation made by the authors, because the points in figure 5 appear very scattered and there is no clear differentiation between groups. This is in agreement with the fact that there is no significant differences in the proportion of FA between groups, at commented before.

Lines 279-283 and Figure 6: I do not understand what is represented in figure 7. Authors should explain it better. On the other hand, author must give the discriminant equation, especially, which FA are included in it.

Answer –  Thank you for your suggestion we have placed the above figures next to each other  (new Figure 5 and 7 ) and made a new figure (Fig 6) of the discriminant analysis

In my opinion, PCA and LDA are two statistical analyses that have similar objectives. Therefore, including both is redundant. Therefore, authors should include only the one that gives the clearest results.

Answer – Both techniques are projecting the data onto a smaller feature subspace: with PCA, we would find the directions (components) that maximize the variance in the dataset (without considering the class labels), and with LDA we would have the components that maximize the between-class separation. We have found that most papers display both PCA and LDA. We believe that it is worth presenting the results of both methods.

Lines 305-309: As commented before, differences in proportion of individual FA and FA groups between mother groups can be stated only if the proper statistical analysis has been carried out, and SD values have to be added. They cannot be inferred from the PCA.

Answer – we have performed the statistical analysis and modified the description accordingly.

Figures 7 and 8 should be put together, as figures Xa and Xb. In this way, as commented before, it could be observed which FAs are characteristic of each group.

Answer –  Thank you for your suggestion we have placed the above figures next to each other  (new Figure 7 )

Discussion

Line 328:  “We observed a decrease in medium-chain fatty acids (capric, lauric and myristic acid) and an increase in oleic acid in the O and O+GD milk samples”. This has not be reported in results, no statistical differences were reported among groups for these FA.

Answer –  The discussion part has been thoroughly reworked

We hope that this improved version of our manuscript will be acceptable for publication.

Reviewer 2 Report

After reading the authors reply and the modified text I consider the manuscript is now adequate for publication.

Author Response

Answers to Reviewer 2’s comments

Thank you for your positive feedback on our revised manuscript.

Thanks again for your valuable earlier comments that helped us improve our manuscript.

We hope that our revised manuscript will be acceptable for publication.

Reviewer 3 Report

The methods used in work were improved. It can be concluded that the method of presenting the results is the only possible with such a limited number of respondents. However, the readability of the charts is limited, and their dimensions should probably be increased. I admire the authors for their diligence and enthusiasm.

Author Response

Answers to Reviewer 3’s comments

 Thank you for your comments and suggestions. We will improve the figures for better readability.

We hope that our revised manuscript will be acceptable for publication.

Round 2

Reviewer 1 Report

In my opinion, the authors have improved the manuscript to some extent, but it still has some major drawbacks that need to be corrected.

Main questions to resolve:
- Figure 1 and Table 2 are redundant. In my opinion, the authors should keep Table 2 and delete Figure 1.
- Statistical differences should be indicated in graphs.
- Although they are different analyses, PCA and Discriminant Analysis give very similar results and, in my opinion, are redundant. They would have to choose one of them.

- The discussion is still very poor. They insist on repeating their results without hardly providing results from the literature, comparing them, or giving possible reasons (metabolic, physiological...) for the results they have obtained.

More especific comments:

Table 1

If I understand correctly the text it should be

Maternal BMI at delivery   23.4 ± 0.3a   23.1 ± 0.4a     31.5 ± 0.6b   32.5 ± 0.5c

Significant differences in "Gender of neonate" are not properly indicated in the table

Paragraph 212-220

Authors must choose between Figure 1 or Table 2. In my opinion, it is better to mantain the table because it includes more FA and the statistical analysis.

If the table is chosen, then it is not necessary to include the percentages in the text because they can be seen in the table.

Table 2,

The row “all samples” does not have much sense, because  the mean of data should not be done when there are significant differences between groups.

For C10:0 and C18:0,  no superscripts are needed if there is no difference between groups.

Why was the statistical analysis of ΣSFA not included?

Paragraph 218-220:  According to data in Table 2, it should be said  something like: “The content of C12:0 was significantly (P<0.05) lower in O+GD group than in the other groups, the content of C14:0 was lower in women with normal BMI than in obese women, and C16:0 was lower in nBMI than in the other groups”.

Paragraph 233-238: Are there statistical differences between the groups? Please indicate it even if there aren't.

Lines 250-251: the difference should be indicated in Figure 2. There are many ways to do so, for example, adding superscripts

Paragraph 263-274:  All data must be mean ± SEM, especially when significant differences were found.  It is not necessary to indicate agian that caproic acid is C6:0, because has been indicated before. And the same applied to other fatty acids.

Figure 4

In my opinion, the graph must include all FA analized (including short chain FA), even though they are in low proportions, and significant differences must be indicated.

Lines 319-322: This is true only for normal BMI mothers, not for obese mothers.

On the other hand, as I commented in the previous review, PCA and discriminant analysis give very similar results. Because of that, in my opinion, are redundant and authors should choose between them.

Lines 397-398: The first sentence does not agree with the following statement.

….comparing breast milk samples from two countries,…….. the goal was to close the knowledge gap regarding how fatty acid composition  in different BMI groups (normal BMI, obese and both combined with gestational diabetes)    (??????)

Paragraph 401-406. In the discussion is not necessary to repeat agian the results, or give them in other forms. In my opinion, this paragraph shoul be in result section if it contains data not already given

Lines 417-419: Are these data from literature?

Other minor comments are marked in the manuscript.

Author Response

Thank you for your useful comments. We improved our manuscript taking into account all your suggestions.

We have inserted our answers below your comments

Comments and Suggestions for Authors

In my opinion, the authors have improved the manuscript to some extent, but it still has some major drawbacks that need to be corrected.

Main questions to resolve:
- Figure 1 and Table 2 are redundant. In my opinion, the authors should keep Table 2 and delete Figure 1.

Answer – Figure 1 was taken out

- Statistical differences should be indicated in graphs

Answer – Statistical differences are indicated in graphs

- Although they are different analyses, PCA and Discriminant Analysis give very similar results and, in my opinion, are redundant. They would have to choose one of them.

Answer –  The results presentation has been modified as requested.

- The discussion is still very poor. They insist on repeating their results without hardly providing results from the literature, comparing them, or giving possible reasons (metabolic, physiological...) for the results they have obtained.

Answer – Discussion was improved with more information

Please keep in mind that the goal of international research is to gain an understanding of the fatty acid composition of mothers with various health conditions. We intend to investigate a deeper biochemical relationship in the future. This paper was prepare upon request for the special issue. To develop the topic, we cited 80 papers. I hope the revised manuscript is acceptable to you as well.

More specific comments:

Table 1

If I understand correctly the text it should be

Maternal BMI at delivery   23.4 ± 0.3a   23.1 ± 0.4a     31.5 ± 0.6b   32.5 ± 0.5c

Significant differences in "Gender of neonate" are not properly indicated in the table

Answer – significant differences are indicated by different letters

Paragraph 212-220

Authors must choose between Figure 1 or Table 2. In my opinion, it is better to maintain the table because it includes more FA and the statistical analysis.

Answer – Figure 1 was taken out, we maintain Table 2

If the table is chosen, then it is not necessary to include the percentages in the text because they can be seen in the table.

Answer – the percentages were deleted

Table 2,

The row “all samples” does not have much sense, because  the mean of data should not be done when there are significant differences between groups.

For C10:0 and C18:0,  no superscripts are needed if there is no difference between groups.

Why was the statistical analysis of ΣSFA not included?

Answer – we made the corrections and additions as suggested

Paragraph 218-220:  According to data in Table 2, it should be said  something like: “The content of C12:0 was significantly (P<0.05) lower in O+GD group than in the other groups, the content of C14:0 was lower in women with normal BMI than in obese women, and C16:0 was lower in nBMI than in the other groups”.

Answer – The suggested sentence was used

Paragraph 233-238: Are there statistical differences between the groups? Please indicate it even if there aren't.

Answer – The MUFA content of the groups did not differ significantly (this sentence was added).

Lines 250-251: the difference should be indicated in Figure 2. There are many ways to do so, for example, adding superscripts

Answer – the significant differences are shown in each figure

Paragraph 263-274:  All data must be mean ± SEM, especially when significant differences were found.  It is not necessary to indicate again that caproic acid is C6:0, because has been indicated before. And the same applied to other fatty acids.

Answer – we made the corrections as suggested

Figure 4

In my opinion, the graph must include all FA analized (including short chain FA), even though they are in low proportions, and significant differences must be indicated.

Answer – short chain FA was included; significant differences are shown in the figure

 Lines 319-322: This is true only for normal BMI mothers, not for obese mothers.

Answer – corrected

On the other hand, as I commented in the previous review, PCA and discriminant analysis give very similar results. Because of that, in my opinion, are redundant and authors should choose between them.

Answer – The results presentation has been modified as requested.

 Lines 397-398: The first sentence does not agree with the following statement.

….comparing breast milk samples from two countries,…….. the goal was to close the knowledge gap regarding how fatty acid composition  in different BMI groups (normal BMI, obese and both combined with gestational diabetes)    (??????)

Answer – corrected

Paragraph 401-406. In the discussion is not necessary to repeat again the results, or give them in other forms. In my opinion, this paragraph should be in result section if it contains data not already given

Answer – the paragraph containing the results has been removed

Lines 417-419: Are these data from literature?

Answer – The missing sentences was included

Other minor comments are marked in the manuscript.

Answer – we have corrected the manuscript taking into account all your suggestions

Round 3

Reviewer 1 Report

The manuscript has improved and could be accepted for publication.
English could improve in some paragraphs.
In the manuscript I have marked some comments.
The main one is that in the materials and methods section, in the description of the statistical analysis, the description of the PCA must be changed.

Author Response

Dear Reviewer,

We appreciate your helpful suggestions for improving our manuscript. Your advice was extremely helpful, and we will use it again when preparing the next manuscript.

We corrected everything based on your suggestions.

Line 188-190

…with random leave-one-out (LOO) cross-validation for the samples from Hungarian and Ukrainian mothers with normal BMI.

Line 311

three fatty acids

Line 342-344

In our study, we compared breast milk samples from two countries, Ukraine and Hungary, in the normal BMI group, to show how pregnant women's fatty acid composition affects the fatty acid composition of breast milk during lactation.

Line 356

European range. These differences

Line 371-372

Obese patients have increased

Line 388

.. between groups.

Line 433

mother’s milk arachidic,

Line 443

oleic,

Line 447

in milk from obese

We are hopeful that the revised manuscript will be accepted for publication.

Prof. Dr Livia Simon Sarkadi

This manuscript is a resubmission of an earlier submission. The following is a list of the peer review reports and author responses from that submission.

Round 1

Reviewer 1 Report

The manuscript describes the fatty acid profil of breast milk from mothers with normal weight, obesity or gestational diabetes.

In my opinion, the work lacks of originality, as many articles and reviews dealt already with this subject, as authors declare in the “Discussion” section. Perhaps there are fewer articles where the comparison between these three mother groups is made. However, the number of participants and the method used to analyze fatty acids, among other factors, make the work presented to have very low scientific significance.

One of the main drawbacks of the work is that it is based on a single analysis of the samples, which is the analysis of fatty acids. And yet, the method they have chosen to analyze them does not allow to accurately determine the fatty acid profile.

It is well known that the FA profile of milk fat is very complex. More than 400 different FAs have been described in some cases, although around 50 FAs can be detected with the commonly used methods.

The method used in the work does not allow to differenciate, e.g., between cis and trans isomers, or to detect CLA. This is a great drawback, because the differences in these minority FAs may be of more interest, in relation to health effects, than the small differences found in the major FAs.

Besides, as I said before, the number of participants in the study is very small and is not not clearly stated which statistical analysis they have applied.

The discussion of the results is very poor. The authors comment on the previously published results and compare them with their own, but do not formulate hypotheses about the possible reason for the differences found.

More specific comments

Isn't it mandatory to include line numbers for easy reference to the text when including a comment?

Since the manuscript does not have line-numbers, I have highlighted the paragraphs in the manuscript and given them a number to refer to in the review.

In general, the ideas in the introduction are included in a messy way and include general knowledge concepts that, in my opinion, should be removed.

More specific comments for the iIntroduction:

  1. very basic knowledge, that, in my opinion, it is not necessary to include

  1. First, these statements need bibliographic references.

In addition, not all trans fatty acids are associated with adverse health effects. It is important to differentiate between trans fatty acids of industrial origin and those of ruminant origin.

In any case, it would increase the level of LDL-cholesterol and decrease HDL-cholesterol.

As far as I know, in the European Union there is no Legislation on TFA limits in food, but maximum intake recommendations. If the authors are aware of the legislation, they should include the reference.

  1. Also basic knowledge, that, in my opinion, it is not necessary to include

  1. All this fatty acids are present in milk in concentration higher than trans Because of that, they should be mentioned before

  1. This paragraph should go after reference [9]

  1. Once the abbreviations have been defined, they should be used. Please, review all the manuscript.

  1. In my opinion, this paragraph should go before talking about GD.

Materials and Methods

  1. Fifteen mothers in total or in each group? If in total, in my opinion, the size of the sample is very small.

  1. Approval of an Ethical Committee?

  1. Is the method based on a method published in the literature? If so, please cite it. Otherwise, the optimization of the method must be described, giving data such as recovery, precision, detection limits... On the other hand, they have to explain how they do the quantification.

Results

  1. In my opinion, the y-axis label is not correct in all figures. The authors don't describe how they quantify FA, but if it's in percent by weight, it should be g FA/100g total FA, e.g. If I understand correctly, each bar corresponds to the mean value measured for each FA in each mother group. If so, they should also include the SD.

  1. Have they carried out the statistical study, for example ANOVA and Bonferroni test, to determine if there are significant differences between groups? This analysis is mandatory and the results have to be indicated in all figures, if there are significant differences between groups. If there are not, then they cannot be cited in the text.

  1. Vaccenic acid is trans11-18:1 (or trans18:1n-7). I have my doubts that the method used allows them to differentiate between cis and trans In any case, and if so, why don't they refer which is the isomer of all unsaturated FA detected?

  1. In M& M, in the statistical analysis section, they refer to a PCA, whose results are not shown.

Other, more specific comments, are included in the manuscript.

Reviewer 2 Report

Fatty acid composition of milk from mothers with normal weight, obesity or gestational diabetes

The topic covered in this paper is relevant to the mother-baby health, linking the maternal nutritional status to the human milk fatty acids composition, which may influence the baby’s health.

Abstract: it is well written and includes the main results of the paper. In order to improve the writing, I suggest including the sample size, as well as the main statistically significant results for the groups comparisons.

Introduction/aim: the introduction contextualizes the topic properly. However, it could be shortened by connecting some sentences and summarizing information.

In “Numerous studies have shown that the mothers’ diet and health can greatly affect the fatty acid composition of breast milk” – there was no reference for this sentence.

 In addition, what is innovative in this paper, since previous ones have already reported the influence of the maternal BMI and Gestational Diabetes on the human milk composition (1,2)?

Methods:

Human milk samples: information about the sample size (n) within each group (nBMI; O; nBMI+GD and O+GD) is missing, please include it.

“Following collection, milk samples were immediately placed in a freezer at –80°C” – wouldn’t be -20°C (usual home freezer temperature)?

in which period at postpartum were the samples collected? This information is missing in the text.

Statistical Analysis: there was no description about the statistic tests that were used to compare the groups. I did not find any principal component analysis (PCA) in results, although it was mentioned in the Statistical Analysis section.

Results:

The results may be compromised by the small total sample size (n =15) split in 4 different groups (with unknown sample size).

Figures: What is the meaning of the y axis in Figure 1/2/3/4? Are the y axes the “% of samples” in which the evaluated fatty acids were identified, such a presence/absence analysis?

Which statistic method was used to make the follow statements, cited in the manuscript:

“Palmitic acid content in breast milk was slightly increased in the nBMI+GD, O and O+GD groups compared to the nBMI group”

or

“Content of MUFAs was slightly increased in samples from the nBMI+GD, O and O+GD groups compared to those from the nBMI group, except for eicosenoic acid for which the content was comparable in all four groups”

or

“The n-3 PUFA content of breast milk from the O group was the highest compared to samples from the other three groups, except for docosahexaenoic acid.”

There are more statements about “differences” among the groups, however I couldn’t see any mention about the statistic method used or the p- value for such comparisons.

Please, show the sample size for each evaluated group also in Figures legends.

I would suggest including a descriptive table to compare the maternal characteristics among the groups. Information about gestational age at delivery, days/months after delivery which the samples were collected are relevant to the findings.

Discussion:

The discussion is based on the results which is compromised by the groups sample size, as reported above.

I strongly recommend rethinking the groups stratification or complement the analysis with more samples, as well as apply appropriate statistical tests to compare the fatty acids among the groups.

References

  1. de la Garza Puentes A, Martí Alemany A, Chisaguano AM, et al. The Effect of Maternal Obesity on Breast Milk Fatty Acids and Its Association with Infant Growth and Cognition-The PREOBE Follow-Up. Nutrients. 2019;11(9):2154. Published 2019 Sep 9. doi:10.3390/nu11092154
  2. Peila C, Gazzolo D, Bertino E, Cresi F, Coscia A. Influence of Diabetes during Pregnancy on Human Milk Composition. Nutrients. 2020;12(1):185. Published 2020 Jan 9. doi:10.3390/nu12010185

Reviewer 3 Report

The title of the work is very interesting.
Unfortunately, the number of breastfeeding women studied is very limited.
I did not notice the numerical division into the study groups.
I did not notice the use of statistical tests to differentiate the results between the studied groups, which is probably justified by the small number of respondents in the given groups.
How can conclusions be drawn from such a small number of respondents?
The introduction of a group of Hungarian women increased one of the study groups, but did it have an impact on the quality of work?
Charts are based on percentages. Can you perform a statistical analysis based on numerical values?

I am sorry, but the work as it stands does little to understand how the different forms of nursing women and the fatty acid composition of human milk affect the composition of milk.